# Colloidal transport by light induced gradients of active pressure

Nicola Pellicciotta [1,2] ✉, Matteo Paoluzzi[3], Dario Buonomo [1], Giacomo Frangipane [1,2], Luca Angelani [1,4] & Roberto Di Leonardo [1,2] ✉

Active fluids, like all other fluids, exert mechanical pressure on confining walls. Unlike equilibrium, this pressure is generally not a function of the fluid state in the bulk and displays some peculiar properties. For example, when activity is not uniform, fluid regions with different activity may exert different pressures on the container walls but they can coexist side by side in mechanical equilibrium. Here we show that by spatially modulating bacterial motility with light, we can generate active pressure gradients capable of transporting passive probe particles in controlled directions. Although bacteria swim faster in the brighter side, we find that bacteria in the dark side apply a stronger pressure resulting in a net drift motion that points away from the low activity region. Using a combination of experiments and numerical simulations, we show that this drift originates mainly from an interaction pressure term that builds up due to the compression exerted by a layer of polarized cells surrounding the slow region. In addition to providing new insights into the generalization of pressure for interacting systems with non-uniform activity, our results demonstrate the possibility of exploiting active pressure for the controlled transport of microscopic objects.

An ideal gas of non interacting self-propelled particles moving in an inertialess microenvironment, will exert a reaction force over a confining boundary that must push on them to overcome self-propulsion and avoid crossing[1]. This resulting "swim pressure" will depend on the number of particles pushing on a unit surface times the mean force applied by them. The surface density of this layer of polarized particles can be estimated by a simple scaling argument[2] that imposes a balance between the rate of particles colliding with a unit surface $-\rho v$ and the rate of particles leaving the surface $-\sigma/\tau$ where $\rho$ is the bulk number density, $v$ is the self-propulsion speed, $\sigma$ the surface density and $\tau$ the average interaction time between particle and surface. By equating the two rates we get $\sigma \sim \rho v \tau$ that, when multiplied by the "swim force"[3] $f \sim \gamma v$ (with $\gamma$ the viscous friction on the particle), finally gives for the pressure $P \sim \gamma \rho v^2 \tau$. A more rigorous treatment shows that the previous expression provides an equation of state for the "swim pressure" of

spherical particles (no torque interactions) with $\tau$ the inverse tumbling rate $\alpha$[1,4–6]. When repulsive interactions are introduced, the motions of the particles are slowed down by their neighbors, causing a reduction in the "swim pressure"[7]. At the same time, a second pressure term appears, which is the ordinary interaction pressure of passive systems[1,7,8] and that becomes predominant at high densities. Again, if particle reorient freely, the total pressure on the boundary can be expressed by an equation of state depending solely on bulk fluid properties[7]. These results have been discussed mainly in simulations but only in few experiments with Janus particles confined by acoustic traps or gravity[9,10]. When active particles are reoriented by walls their mechanical action on confining walls cannot be deduced any longer from a state function depending solely on bulk properties[4,11], as also verified experimentally in granular active systems[12]. Even in those simpler cases when active pressure is a state function, some apparently

[1]Dipartimento di Fisica, Sapienza Università di Roma, Piazzale A. Moro 5, 00185 Roma, Italy. [2]NANOTEC-CNR, Soft and Living Matter Laboratory, Institute of Nanotechnology, Piazzale A. Moro 5, 00185 Roma, Italy. [3]Departament de Física de la Matèria Condensada, Universitat de Barcelona, C. Martí Franquès 1, 08028 Barcelona, Spain. [4]ISC-CNR, Institute for Complex Systems, Piazzale A. Moro 5, 00185 Roma, Italy. ✉e-mail: npellicciotta@gmail.com; roberto.dileonardo@uniroma1.it

odd mechanical properties emerge when activity is not spatially homogeneous. In passive fluids, the mechanical equilibrium condition requires that, in the presence of a non-homogeneous temperature, a modulation of density is generated to restore uniform pressure throughout the fluid. For a low density active fluid, instead, when the swimming speed varies in space, the stationary density satisfies $\rho v \simeq$ const[13–15] resulting in a non uniform pressure $P \propto v^4$. Thus regions with different activities coexist side by side in mechanical equilibrium as long as the product $\rho v$ is constant. Although in equilibrium with each other in the bulk, regions with different activities exert different pressures on the walls with the faster region pushing harder[4]. This remains true even when repulsive interactions between active particles are introduced, although the pressure difference on the two sides can be significantly reduced by the direct pressure term on the slow, more concentrated region[16].

Here we show that a stationary active fluid with a space-dependent activity conceals pressure gradients that become manifest when an external object is immersed in the fluid and experiences unbalanced pressures on its surface. By using *Escherichia coli* cells genetically engineered to swim with a light controllable speed[14,17,18] we continuously keep a spherical probe in between a high- and a low-speed regions by projecting dynamic light patterns that follow the probe movements. At high enough densities of bacteria, we find that the probes are pushed stronger by the low-activity side, in contrast to what expected from previous simulations where a higher pressure was found on the wall facing faster bacteria[7,16]. Although the transport of asymmetrical objects in homogeneous bacterial baths has already been demonstrated[19–23], here we show that the requirement of spatial symmetry breaking, which is necessary for rectification, can be transferred from the object to the fluid, enabling the active transport of passive objects with arbitrary shapes. Using a combination of experiments and numerical simulations, we show that the higher pressure pushing the probe from the low activity side is indeed a passive interaction pressure originating from the compression produced on the slow region by a surrounding layer of polarized cells.

## Results

### Observation of colloidal drift in light-induced activity gradients

Our active fluid consists of a suspension of *E. coli* bacteria expressing the light driven proton pump proteorhodopsin. In the absence of oxygen, aerobic respiration is blocked and proteorhodopsin can be used to control the proton motive force that drives flagellar motors, so that the bacteria's speed becomes a function of local green light intensity[17]. The response of our strain to green light intensity is reported in Supplementary Fig. 1. Using a digital light projector coupled to the microscope objective, we can shape the speed of bacteria with a spatial resolution that matches that of the imaging system[14,24]. A space-dependent speed gives rise to a density modulation that, in the steady-state, accumulates bacteria in low-speed regions and depletes them from high-speed regions. Static modulations have been used to shape bacterial density with light[14,18]. Dynamic modulations instead can be used to generate steady fluxes of bacteria[25,26] or to trap dense and active clouds of bacteria using an optical feedback loop[24]. Here we are interested in the mechanical effects of a space modulated activity on suspended passive objects. For simplicity we choose spherical probe particles (polystyrene 5 µm radius) so that any observed rectification must be attributed to a broken spatial symmetry in the bacterial bath. In a series of preliminary experiments, we produced smooth speed gradients on length scales larger than the particle size, but no significant drift was observed, Supplementary Fig. 2. Therefore we decided to expose the particle to a steep activity modulation that is found at the interface between a low speed region (light intensity $I_-$ and corresponding mean bacterial speed $v_-$) and a high speed region ($I_+$, $v_+$), Fig. 1a. We project this binary pattern within circular regions of radius $R$ whose centers are continuously updated to follow the position of the probe particles. Outside these modulation disks we set a uniform background of intensity $I_0$ corresponding to a mean speed $v_0$. In this way we can record multiple independent particle trajectories in the same field of view. We image the system in dark field mode so that bacteria appear as white spots over a dark background while the colloidal probes stand out in the image as white saturated circles, Fig. 1b. When modulation circles are projected, bacteria start accumulating in the low speed side producing a brighter half disc on the left side of the probes. This transient dynamics lasts for a few seconds (see Supplementary Fig. 3) during which the probe settles into a final systematic drift to the right. Particle trajectories spanning a total time of 5 min are superimposed in green over the last image frame in Fig. 1b, c (See Supplementary Movie 1). The drift direction is reversed when we switch high- and low-speed regions within the modulation circles. We measured drift speeds of V ≈ 10 µm/min, with light intensities of $I_+ = 0.9$ mW/mm², $I_- = 0.1$ mW/mm² and $I_0 = 0.7$ mW/mm² ($v_+ = 19$ µm/s, $v_- = 7$ µm/s, $v_0 = 17$ µm/s), in a modulation circle of radius $R = 40$ µm.

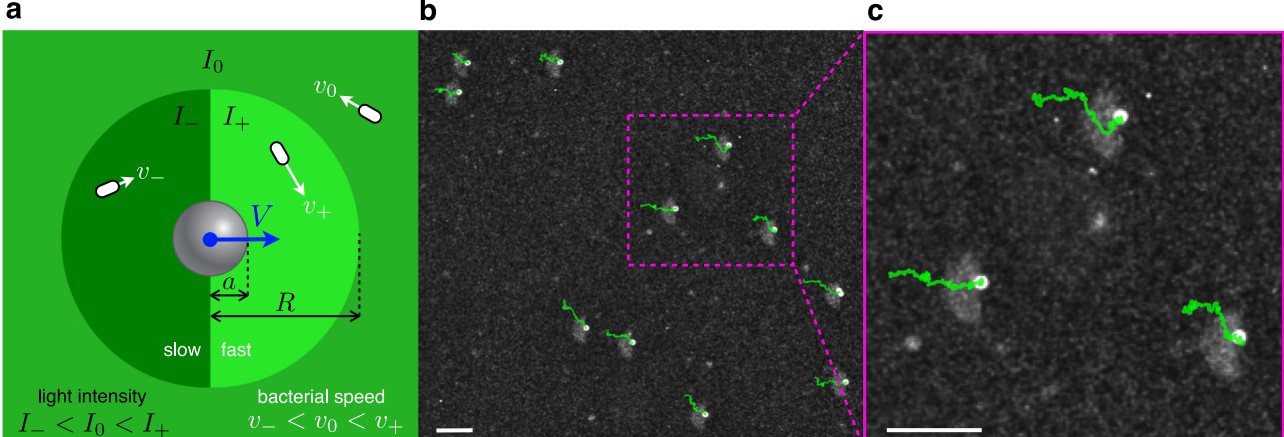

**Fig. 1 | Colloidal transport in light-induced activity gradients. a** We project patterns of green light to modulate bacterial activity around colloidal particles. A modulation disc of radius $R$ is centered on each particle and divided into two halves with light intensity values $I_- < I_+$ and corresponding velocities $v_- < v_+$. Outside these modulation disks we set a uniform background of intensity $I_0$ corresponding to a mean speed $v_0$. The position of these disks is continuously updated to follow particle positions. **b** A sample snapshot of the system taken using dark-field microscopy shows several colloidal beads and their trajectories (in green). When illuminated with the structured pattern in **a**, bacteria accumulate in the slow side and the colloidal beads drift systematically in the direction pointing towards the fast side. **c** A magnified view of the box shown in **b**. Parameters: $R = 40$ µm, $I_- = 0.1$ mW/mm², $I_0 = 0.7$ mW/mm², $I_+ = 0.9$ mW/mm². Scale bars are 50 µm.

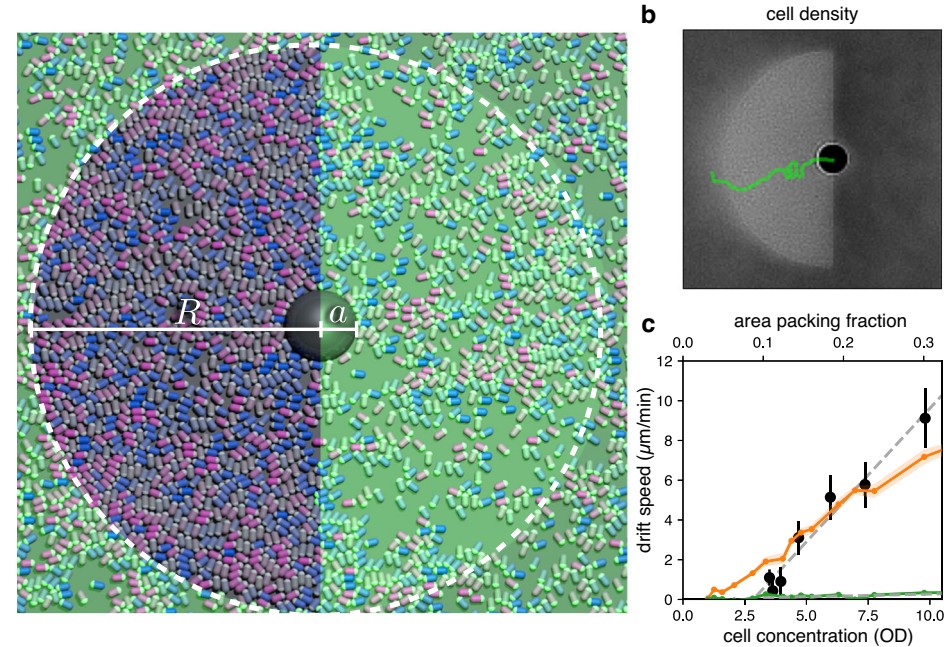

**Fig. 2 | Bacterial dynamics simulations. a** A snapshot of a 2D numerical simulation reproducing the experimental design in Fig. 1. A passive particle always lies at the interface between two regions where bacteria have different speeds, $v_+ > v_-$ while for distances greater than $R$ from the bead's center, bacteria move with speed $v_0$. Blue to pink colormap encodes bacteria orientation. **b** Simulated beads also drift in the direction pointing from slow to fast bacteria. A simulated bead trajectory is reported in green over an average density map of bacteria. **c** Colloidal drift increases almost linearly with cell density in both experiments (black dots) and

simulations (orange line) both extrapolating to zero drift for a finite density value. Turning off steric interactions between bacteria the drift is significantly reduced and becomes proportional to density (green line). We scaled the area packing fraction axis in simulation data so that 0.38 corresponds to a cell concentration of 13.5 optical density (OD), see Supplementary Notes 1 and 2. Experimental points are obtained as averages over more than 50 particles. For simulations, we first calculate time averages over $10^6$ steps and then average over 20 independent runs. Error bars are the standard error of the mean.

This drift is about 1% of bacteria speed in the fast side. Contrary to our expectations, however, the movement of the microspheres is directed towards the high-speed side, as if the pressure exerted by the slow bacteria was greater than that of the fast ones. As discussed in the "Introduction", previous simulation work found a higher "swim pressure" coming from the more active right side, which could be only partially counterbalanced by a direct interaction pressure term if the slow side becomes sufficiently concentrated[16].

To get some clues on the physical origin of this drift, we first analyze its dependence on the bulk bacterial concentration. As shown in Fig. 2c, a significant drift is only found for optical density (OD) values above 4 with a best fit line going to zero for a finite OD. This observation suggests that interactions between bacteria play an important role and drift is not simply proportional to density as in the case of non interacting cells. This finding is also confirmed by bacterial dynamics simulations shown in Fig. 2, see also Supplementary Movie 2. Steric interactions between bacteria can be easily turned off in simulations resulting in a strong suppression of drift (that anyways maintains a positive sign). On the other hand, when we consider steric interactions between swimmers, we find a good quantitative agreement with the experimental data Fig. 2c. Although interactions can also be hydrodynamic in principle, we believe that these play a minor role in our experiments and therefore we do not explicitly consider fluid in our simulations. In fact, hydrodynamic interactions in a force-free system occur only through dipole or higher order multipoles, becoming effective only over short distances where steric interactions are the predominant mechanism for alignment[27]. Finally, in these experiments bacteria are either highly motile and low concentrated or highly concentrated and non-motile, another reason to justify neglecting of hydrodynamic interactions.

## Shaping active pressure with light

In passive particle systems, the mechanical pressure acting on a solid wall $p_W$ is composed of two contributions: a kinetic (or osmotic) pressure term $p_T$ plus a direct (or virial) pressure $p_D$ originating from interparticle forces. When activity is introduced $p_T$ becomes negligible and it's replaced by a new term, $p_S$, the swim pressure arising from the propulsion forces of active particles that are in contact with the wall and oriented against it[1]. Activity also indirectly affects $p_D$ by contributing to determine the density correlators that appear in the definition of $p_D$[7]. In the idealized situation where the reorientation dynamics of active particles is not affected by interactions with wall or other particles, it is found that $p_S$ can be expressed as a state function of bulk variables[7,8]. In both our experiments and simulations, interactions between bacteria and with the probe bead generate external torques that reorient active particles. Nevertheless, we can always calculate $p_D$ throughout the simulated system and evaluate $p_W$ over the surface of the bead. Figure 3b shows the mechanical pressure $p_W$ acting over the contour of the probe bead for three different positions of the bead with respect to the modulation mask. When the bead is fully immersed in the fast half disc, a uniform pressure is measured over the contour with a mean value given by $p_W = 1.1$ in simulation reduced units (see Supplementary Note 2). When the particle is fully embedded in the slow region, still an approximately uniform pressure is found but this time with a much larger mean value of $p_W = 2.1$. Supplementary Movies 2–4 show the dynamics of the simulated particle in the three positions. We can now compare these values of $p_W$ with the direct pressure $p_D$ evaluated in the neighborhood of the bead. We first calculate full $p_D$ maps as reported in Fig. 3a and then extract $p_D$ values around a circular contour right outside of the bead's surface Fig. 3b. When the particle is in the slow region we find that the mechanical pressure $p_W$ can be fully explained by the direct pressure term $p_D$. We

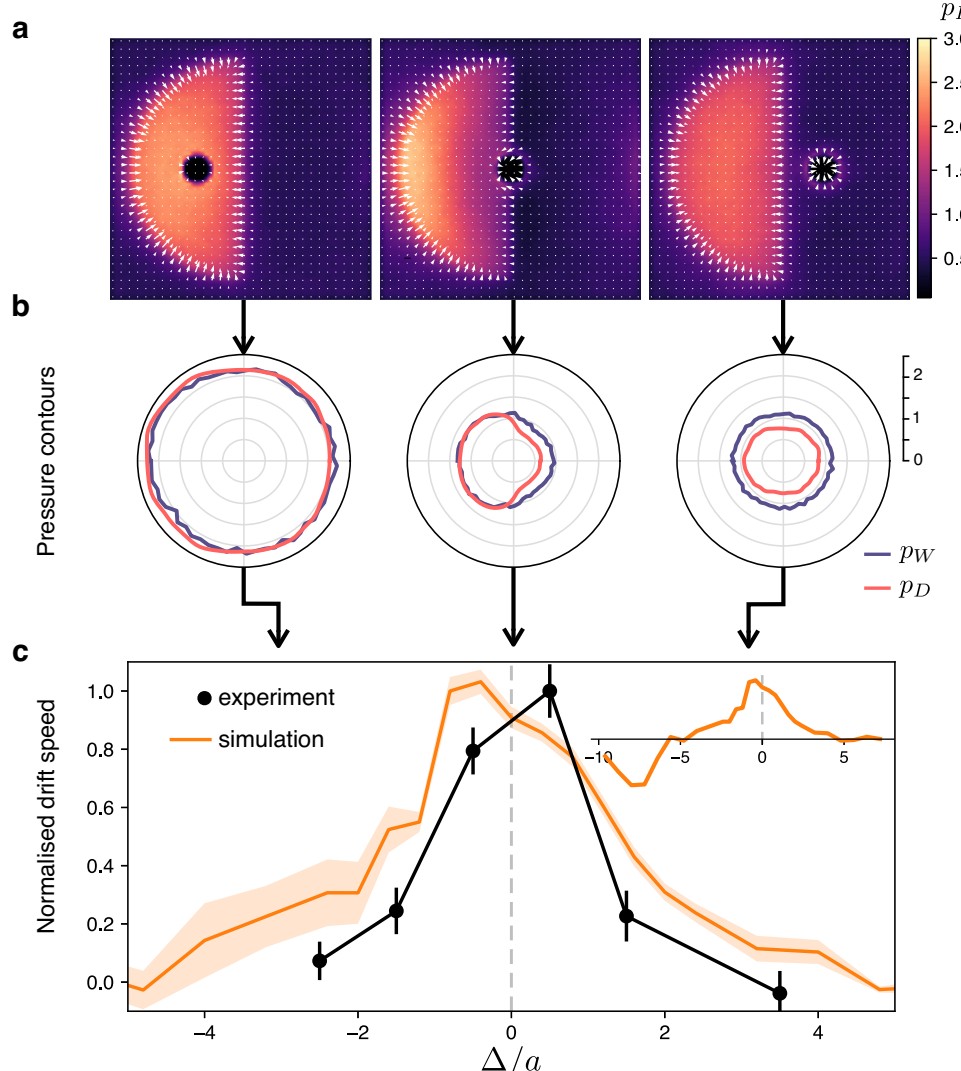

**Fig. 3 | An active pressure imbalance drives particle drift. a** Direct (interaction) pressure maps obtained from simulations in which the bead is fully embedded in the slow region (left panel), in the fast region (right panel), and at the interface between them (central panel). The swim force density $\mathbf{f}_S(\mathbf{r})$ is superimposed on the maps as the white vector field. **b** Polar plots of the mechanical pressure $p_W$ and the direct pressure $p_D$ acting along the bead boundary for the three different configurations in **a**. **c** Drift speed as a function of the displacement $\Delta$ from the center of the modulation disc. The drift is maximum when the bead is on the interface and vanishes as the bead moves inside the slow or the fast regions ($|\Delta/a| > 1$). The inset shows simulated drifts over a wider range of $\Delta$ values. A negative drift appears when the bead position is at the left interface between the slow region and the background ($\Delta/a = -8$). Experimental points (in black) are obtained as averages over more than 50 beads. Simulation points (orange line) are obtained by averaging over 6 independent runs with $10^6$ steps. Error bars represent the standard error of the mean.

can empirically define a swim pressure as the difference $p_S = p_W - p_D$ and conclude that it is negligible in the slow region. When an equation of state exists, $p_S$ is found to be proportional to the product of the free self-propulsion speed and the actual mean particle speed. Both these speeds are slow in the dark half disc because of the low activity induced by $I_-$ and the slowdown caused by the closely packed neighbors. On the contrary, when the bead is in the fast region, $p_D$ only accounts for 70% of the mean $p_W = 1.1$ with an extra pressure of 0.3 that can be attributed to $p_S$. In both these situations the net force on the particle and consequently the drift are both very small. Moving the particle right at the edge between the fast and the slow region $p_W$ becomes unbalanced and a net force appears causing particle drift. Although $p_W$ is found to decrease on both the fast facing half and on the slow one, the pressure on the slow side remains higher and pushes the bead towards the right. Comparison with $p_D$ shows that direct pressure is still responsible for all $p_W$ on the slow side, while $p_S = p_W - p_D$ remains equal to 0.3 as in the case where the bead is

completely immersed in the fast region. It is also interesting to compare this estimated value of $p_S$ to the swim pressure $\gamma\rho v_+^2\tau/2$ of an ideal active gas of run and tumble particles in 2D[1]. Using simulation parameters (in reduced units as defined in the Supplementary Note 2) $\gamma = 1$, $v_+ = 1$, $\tau = 10$ (inverse tumbling rate) and the average value of the particle density in the fast region $\rho = 1$, we find an ideal swim pressure that is greater than $p_S$ by an order of magnitude. This may be due to an effective interaction time $\tau$ that is not controlled by the inverse tumbling rate but rather by the typical interaction time with the bead scaling with $a/v_+ = 2.5$. We can speculate that an interaction time that scales as the inverse speed may also be the reason why the swim pressure $\sim \rho v^2\tau$ is practically constant ($\rho \propto 1/v$, $\tau \propto 1/v$) in the absence of interactions (see Fig. 2c) or at low densities such as those obtained in smooth speed gradients (see Supplementary Fig. 1). Experimentally, we can only access the integrated value of $p_W$ producing the drift. However, if the above interpretation is correct, we should be able to increase the drift speed if we could increase $p_D$ on the slow side leaving

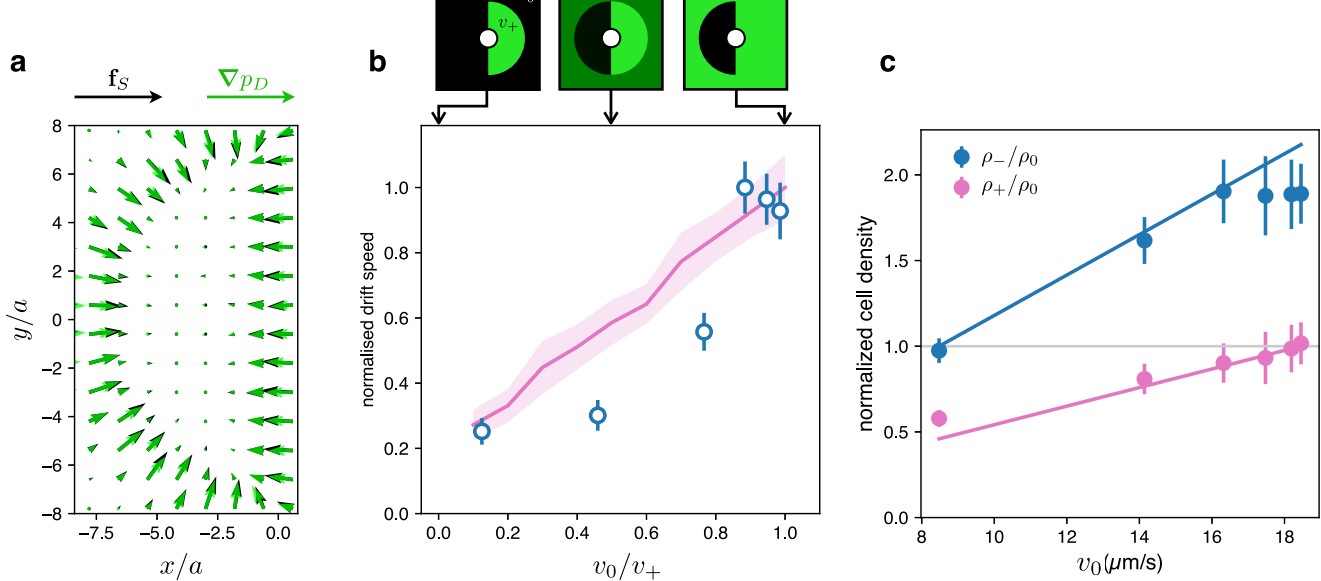

**Fig. 4 | Drift velocity increases with the speed of the background bacteria.** **a** When the bead is not at the interface and currents are absent, the direct pressure gradient $\nabla p_D$ (green arrows), obtained from simulations, is locally balanced by the swim force density $f_S$ (black arrows) as expressed in Eq. (1). **b** If the two speeds $v_-$ and $v_+$ inside the modulation disc are kept fixed ($v_+/v_- \simeq 3$ in this case), but the background speed $v_0$ is increased, the drift speed of the colloidal particle increases both in experiments (blue circles) and simulations (pink line). Drift speeds are normalized to their maximum values. **c** Changing the velocity $v_0$ of bacteria in the

background results in adjustments of the cell density in the three regions. We show experimental values of the ratios $\rho_-/\rho_0$ and $\rho_+/\rho_0$ as a function of $v_0$, with the ideal (non-interacting) trends represented by the two lines with slopes $1/v_-$ and $1/v_+$ respectively. A deviation from the ideal line is observed in the slow, denser region where $\rho_-/\rho_0$ seems to saturate for $v_0 > 16\,\mu\mathrm{m/s}$. Experimental points are obtained as averages over more than 50 beads. Simulation points are obtained by averaging over 20 independent runs with $10^6$ steps. Error bars represent the standard error of the mean.

$p_S$ on the fast side unchanged. It has been pointed out already that, in the absence of particle currents a simple local momentum balance on the active particles must result in[16,28]:

$$\gamma v(\mathbf{r})\mathbf{m}(\mathbf{r}) = \nabla p_D \qquad (1)$$

with $\mathbf{m}(\mathbf{r})$ the polarization field as defined in[16], $\gamma$ the viscous drag along the self-propulsion direction and $v(\mathbf{r})$ the speed pattern imposed with light. Eq. (1) states that the swim force density $f_S(\mathbf{r}) = \gamma v(\mathbf{r})\mathbf{m}(\mathbf{r})$ must locally balance the gradient of direct pressure $p_D$. Figure 3a reports the vector field $f_S(\mathbf{r})$ superimposed to the direct pressure field. Details on the calculation of $f_S(\mathbf{r})$ are given in Supplementary Note 2. It is quite evident that the high $p_D$ value in the slow region is maintained by a layer of polarized cells compressing densely packed cells in the dark half disc. When the probe is fully immersed in one of the two half disks and the current is zero we verify quantitatively Eq. (1) as shown in Fig. 4a. This suggests that if we increase the background speed $v_0$, leaving light intensity within the modulation disc unchanged, $p_D$ in the slow region must increase resulting in a higher drift speed of the probe bead.

Figure 4b shows that, although the velocity gradient at the interface is kept constant (fixed $v_-$ and $v_+$ with $v_+ \sim 3v_-$), the drift velocity strongly depends on the background intensity in both experiments (blue circles) and simulations (pink line). It should be noted however, that, although the speeds in the slow and fast half disks remain constant, changing the background speed $v_0$ results in density readjustments as shown in Fig. 4c. Further details of density measurements in the three regions are shown in Supplementary Fig. 4. Ideally, for non-interacting bacteria, density ratios should match speed ratios with the constraint of conserving the total cell number. Specifically, the densities $\rho_+$ and $\rho_-$, when normalized to the background density $\rho_0$, should increase linearly with $v_0$ with slopes given by $1/v_+$ and $1/v_-$, respectively, shown as solid lines in Fig. 4c. In the dark region, however, $\rho_-$ follows this ideal trend only up to a saturation value that is approximately

twice the background density $\rho_0$ and where steric interactions probably prevent further accumulation. Both this saturation of $\rho_-$ and the associated slowing down of bacteria in densely packed regions should decrease swim pressure on the slow side[7] while a strong direct pressure component builds up due to close packing and produces a faster drift.

## Discussion

Our results can be explained by recognizing that activity can generate mechanical pressure on solid walls in two main ways. The first is the "swim pressure" or the direct thrust applied by a layer of active particles that are directly in contact with the wall and having a direction of self-propulsion polarized towards it. The second is through polarized regions in the bulk acting like self-generated body forces that can compress repulsive particles and generate a strong direct pressure term in response[3,16]. Both of these components can be modulated by shaping motility in space. In particular we find that when a probe particle is placed at the interface between a slow and a fast region, although swim pressure alone should be higher on the fast side, a layer of polarized cells surrounding the slow area generates there a direct pressure component that is strong enough to push the particle towards the fast, higher swim pressure side. Stated this way, the drift should not be strongly influenced by the shape of the modulation boundary and indeed this is confirmed by simulations using a square modulation region with the same area as the disc (see Supplementary Fig. 5). However, changing the area of the modulation region can have strong effects on the drift speed, affecting the density ratios in the three regions with different activities (see Supplementary Fig. 6). With the number of light-activated systems steadily increasing, understanding how a pattern of activity translates into a pressure field may stimulate the development of new transport strategies in closed microfluidic systems where pressure gradients could be generated internally. Future studies may also address the possibility of

generating a net torque on anisotropic and achiral objects suspended in bacterial baths with spatially patterned motility.

## Methods

### Sample preparation

We grow the *E. coli* strain AB1157 transformed with a plasmid encoding the proteorhodopsin (PR), and depleted of the *unc* cluster that codes for $F_1 F_o$-ATPase as described in ref. 24. Then, the cells are washed and suspended at the desired concentration in a motility buffer containing 10 mM potassium phosphate (pH 7.0), 0.1 mM EDTA (pH 7.0), 0.002% Tween 20, and lactate 10 mM. The dense suspension of photokinetic bacteria (optical density OD $\approx$ 15) is mixed with a ratio of 10:1 with sterically stabilized Polystyrene (10 µm PS-PEG microparticles, microParticles GmbH) at concentration 1% solid. The final concentration of particles in the sample is approximately 0.1% solid, and bacteria have an optical density of OD = 13.5. The mixed suspension is then loaded into commercial 3 µL sample chambers with a thickness of 20 µm (Leja, NL). The chamber was previously coated with PLL-PEG (PLL(20)-g[3.5]-PEG(5), SuSoS) using proprietary protocols to decrease the adhesion of the microparticles to the substrate. After cell loading, the chamber is sealed using vaseline to stop airflow. In this way, bacteria stop swimming once dissolved oxygen is exhausted[29]. In our case, this happened after a few minutes. The filled chamber is then placed under the optical microscope and observed in dark field illumination, focusing on the bottom plane of the chamber where the particles sediment.

### Optics

Dark field imaging is performed using a custom inverted optical microscope equipped with a ×4 magnification objective (Nikon; NA = 0.13) and a high-sensitivity CMOS camera (Hamamatsu Orca-Flash 2.8). Patterns of green light (520 nm) are generated using a digital light processing (DLP) projector (Texas Instruments DLP Lightcrafter 4500) coupled to the same microscope objective used for imaging through a dichroic mirror. A more detailed description of the custom setup can be found[14]. The positions of the modulation disks in the the projected light patterns follow particle centers through an automatic feedback algorithm updated every $dt = 0.5$ s.

### Numerical simulations

In the numerical model, swimmers are rod-shaped run-and-tumble particles of length $\ell$ and thickness $\sigma$. We indicate with $v$ their self-propulsion speed and with $\alpha = 1/\tau$ the tumbling rate at which particles undergo random reorientation of swimming direction. Immersed into the active bath there is a colloidal particle of radius $a$. All interactions between bacteria and between bacteria and the passive colloid are modeled by steric repulsive forces (the details about the numerical implementation can be found in Supplementary Note 2 and in refs. 23, 30, 31). To mimic photokinetic bacteria, we consider a swimming speed that varies in space according to light templates. As in the experimental set-up, in the colloidal particle's frame, we individuate a circular region of radius $R$, centered at distance $\Delta$ from the center of the colloid. We consider three regions, the background (outside the circle) where particles move at speed $v_0$, and the two semicircular regions of high $v_+$ and low $v_-$ speed (see Fig. 1a). Simulations are performed in a square box of side $L$ where periodic boundary conditions are employed. Choosing motility parameters suitable for describing *E. coli* bacteria, we set $\alpha = 0.1$ ($\tau = 10$), $\sigma = 1/2$ (in unit of $v_0 = 1$ and $\ell = 1$).

## Data availability

Data supporting the results of this study are available from the corresponding authors upon request.

## Code availability

The code used for the simulation is available from the corresponding authors upon request.

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

## Acknowledgements

We thank C. Maggi and S. Bianchi for useful discussions. The research leading to these results has received funding from the European Research Council under the ERC Grant Agreement No. 834615 (R.D.L.). L.A. and R.D.L. acknowledge financial support from the Italian Ministry of University and Research (MUR) under the PRIN2020 Grant No. 2020PFCXPE. M.P. has received funding from the European Union's Horizon 2020 research and innovation program under the MSCA grant agreement No 801370 and by the Secretary of Universities and Research of the Government of Catalonia through Beatriu de Pinós program Grant No. BP 00088 (2018).

## Author contributions

N.P., G.F., and R.D.L. conceived the original idea and designed the experiments. G.F. performed preliminary experiments. N.P. and D.B. performed the experiments. M.P. and L.A. conceived and performed the numerical simulations. N.P., D.B., M.P., and R.D.L. analyzed the data. All the authors discussed the results and contributed to writing the manuscript.

## Competing interests

The authors declare no competing interests.
