## [Peer Review File · Nature Communications]

Colloidal transport by light-induced gradients of active pressureREVIEWER COMMENTS

Reviewer #1 (Remarks to the Author):

The manuscript presents experimental data clearly demonstrating that passive colloids suspended in an active fluid can be transported by tailored spatial modulation of the swimming speed of the active agents. Specifically, the authors use light patterns projected on dense suspensions of light-controlled *E. coli* bacteria to generate areas of high and low swimming centred on passive tracers. Feedback is used to adjust the light pattern (and thus the swimming speed of the bacteria) such that the tracers stay centred on the interface between high and low swimming speed, leading to the observation of a slow systematic drift of the tracer particles towards the side with higher swim speed. The experimentally found direction of drift is unexpected, as simple scaling arguments for an ideal active gas predict that the 'active pressure' exerted by the swimming bacteria should be higher in the region of higher swim speed. The manuscript then presents numerical simulations which show a drift direction consistent with their experimental findings and try to rationalise this in terms of imbalance found in the swimming force density associated with their imposed spatial activity pattern.

Although the experimental findings are significant and of general interest, the link with active (swim) pressure as presented in this manuscript (and advertised in its title) is not convincing. Indeed, after providing a simple scaling argument that would predict a different direction of motion the authors essentially drop the concept of swim pressure, relying on analysis of swim force density from simulations to add support to their experimental findings. As the authors point out, there is still some ongoing debate about the definition and properties of pressure in active systems and I feel that it is not a useful concept in the context of this experiment. Swim pressure is the pressure needed to confine active particles, i.e. it is analogous to osmotic pressure of passive colloids. While it is straightforward to extract this swim pressure in theoretical treatments by assuming semi-permeable membranes as boundaries, the solid colloid in the scenario presented in this manuscript is not just influenced by the motion of the bacteria but also by the fluid flow set up by the bacteria's motion.

Although the simulations support the direction of motion found in experiments, I am not sure whether they capture the experimental system sufficiently to provide any real insights. To start with the bacterial densities used in simulations seem extremely high (up to 0.38 area fraction?). Typically an OD = 1 sample of *E. coli* cells has about 10^9 cells/ml, i.e. a body volume fraction of 0.2%. So even for the densest experimental sample the (3D) volume fraction would have been below 5%. Could the authors comment? And the snapshot shown in fig.2A suggests that the low intensity region has an even higher density which seems to lead to local alignment and presumably correlated motion. Surely hydrodynamic interactions between swimmers (and the colloid) can no longer be neglected? Fig. 3C also suggests that the drift speed is not so much controlled by the velocity gradient at the interface next to the tracer colloid, but rather by the (curved) interface of the low intensity region. Could the authors comment more why this might be the case? I assume the density in the dark region will be very different for varying 'background' illumination?

The experimental evidence provided in the manuscript is of general interest and seems methodologically sound, but the manuscript would benefit from some more detailed characterisation of their experimental system:

1. *E. coli* strain. It is known that light-controlled bacteria do not react instantly to changes in light intensity. Seeing that bacteria only need a few seconds to traverse different regions even a small delay could have a large experimental impact. Furthermore, on a similar light-control run and tumble strain it has been found that their 'tumbling rate increases noticeably as cells swim from light to dark, whereas cells swimming from dark to light do not show any obvious change in their tumbling behaviour.' (J.Arlt et al., Nat Comm. 10, 1 (2019)), which

could potentially complicate the picture further. Is this also the case here?

2. Density profile: Fig. 1 shows that there is a clearly defined high density region where the light intensity is lower. How does this density compare to a) the initial density, b) the density in the high intensity and background intensity region? Visually it looks much denser than would be expected if $\rho \cdot v$ where constant (which of course is unlikely to apply in this experiment)... And can you observe correlated bacterial motion in these high density regions?

3. How long does it take for such a dense region to be established? The manuscript suggests that the colloids only start moving 'systematically' after this has been established. Does it (on average) stay centred on the interface during this time? And how long does it take the particle to reverse direction after the direction of the pattern has been flipped?

4. Some of the authors have recently demonstrated how the swimming of light controlled bacteria can be rectified by an optical feedback loop (Nat. Comm 13, 1 (2022)). Have the authors considered that something similar might be happening here? E.g. the dense region might follow shifts of the pattern towards the brighter half more quickly than shifts in the opposite direction.

Minor points/corrections:

- Scale bar size in fig. 1b&c not mentioned.
- Eq (1) v_i on LHS should be typeset as vector as well
- Caption fig 2A): Pink and blue bacteria are those swimming toward the regions with higher and lower activity, respectively. Do you really mean that? Not simply pink/blue to positive/negative x direction?
- Fig. 3. A) where are the boundaries of the light region? I.e. is the pink region on the LHS just inside or outside the dark area?
- B) Scale would be useful: at least centre position in x and 0 in y
- Line 179/180: shouldn't this read 'for the integrated positive contribution'?
- Caption supp fig. 1: Hill equation expression wrong.
- Caption supp fig 2. Orange line (not blue line)

Reviewer #3 (Remarks to the Author):

This is a quite novel and well conducted study, which in particular challenges current theoretical predictions of active matter. As a result I can recommend this work for publication in Nature Communications. Nonetheless there are several pending questions that I would like the authors to address.

The full review is available as an attached pdf file.

Review for: Nicola Pellicciotta, Matteo Paoluzzi, Dario Buonomo, Giacomo Frangipane, Luca Angelani, and Roberto Di Leonardo, “Colloidal transport by light induced gradients of active pressure”.

In this manuscript, Pellicciotta et al use light-sensitive motile bacteria to produce spatial variations in bacterial activity. They show that a passive colloid placed in a bacterial bath with spatial variations in activity can achieve net directed transport. The authors write that it is surprising that the colloids move from regions of low to high activity based on arguments of the ideal-gas pressure. Interactions between the swimmers are critical to the observed phenomenon, indicating that the ideal-gas like pressure arguments alone fail to explain the results. Apparently, the crowded bacteria on the less active side act as a connected object that gets pushed collectively from the bacteria in the bulk (background) region. To validate this hypothesis, the authors vary the background activity, and the colloidal transport is maximized at the largest background activity (Fig. 3). Surprisingly, Figure 4 shows that the localization of the colloid is critical – the colloid placed at the interface produces the largest transport, whereas the colloid placed within the interiors of the disk does not.

The experiments are very nice; very creative use of light-induced bacterial activity. Results are also interesting. I am willing to reconsider a revised version of this manuscript, but in the current manuscript, as written, I have a concern with the proposed mechanism and model for the observed phenomena. I do not have an issue with the results/data, but I have an issue with the authors’ interpretation and mechanism.

This review may sound overly critical, but I am actually quite supportive. The experiments are very nice and will add to the body of work in a meaningful way. I genuinely want to help the authors improve this work. I just disagree with the proposed mechanism, and this may require a significant reconstruction of the manuscript.

First, I believe that the way the manuscript is introduced is misleading and unrelated to the major conclusions of this work. In the abstract, the authors bring up the *ideal-gas* active pressure, and that the probe moves from regions of low to high pressure. This is an argument based on dilute, non-interacting systems. However, the main results of this work is entirely reliant upon nonideal interactions between the bacteria. I understand the need to be provocative, especially for high-impact journals like this, but it’s to a degree and at the expense of scientifically misrepresenting the problem. After reading the abstract and the introduction, I had the impression that the bacteria are dilute, and two-body interactions between the bacteria plays no role in this “surprising” result. In reality, the interactions play an essential role in the transport. I still believe that the colloid is going from high to low *total* pressure, which includes the active pressure and interaction pressures. So the results are actually not that surprising to me, when presented in this way.

According to the authors’ proposed mechanism, the whole assembly (colloid + dense bacteria on the less-active hemi-circle) is being transported, as opposed to moving just the colloid. Presumably, one could just place a rigid object in the shape of a hemi-circle, and the hemi-circle would still move. More generally, one could just create any object with sufficient asymmetry and curvature, and the object should move. This fact seems to decrease the impact of this manuscript a little bit, if this mechanism is really true.

If I understood correctly, the authors' proposed mechanism is that effectively the system takes on the shape that looks like the following:

The authors claim that this background activity breaks the symmetry of forces and causes net translation of the object as a whole.

I disagree with this proposed mechanism.

The arguments presented in Eqs (2) – (4) are weak in my opinion. A better approach would be to follow Yan & Brady “The force on a boundary in active matter”, JFM 2015, which is not cited in the current manuscript (to me, this is an important missing reference, given the authors' proposed mechanism). Equation (2.2) in Yan & Brady is the direct calculation of the force on any object, given the number density of particles. In short, this expression states that every particle gives a kT -kick on the boundary, so all we need to do is to integrate the number density of particles around the whole boundary surface. I encourage the authors to calculate this force based on the known density at every point around this whole object in both the experiments and simulations, shown above.

I doubt that the authors will calculate a net positive force towards the right. Yan & Brady 2015 JFM showed that a strong concave curvature is required to achieve sufficient asymmetry in kT -kicks for net directed transport of the whole object. See Fig. 3B in Yan & Brady 2015 JFM – their shape almost looks like the shape drawn above (!), except without the concave curvature. Furthermore, this would argue that the object would go in the opposite direction, towards the left! I find it hard to believe that the authors' presented mechanism would cause the object above to achieve net directed transport towards the right.

For me to believe the authors' proposed mechanism, I would like to see a simulation of a rigid object in the shape of the whole disk object (the shape I drew above), placed inside a bath of active particles swimming at the bulk background activity. If the simulations show that the whole object moves towards the right, I would be convinced (I would be very surprised if this happens!). The only thing I can think of is the shape of the active particles are non-spherical, unlike the spherical particles used in the theory – but I doubt that the small deformation in active particle shape would reverse the direction of net transport.

Of course, the experiments are what they are, and they show that the object moves towards the right. I conclude then that the authors' proposed mechanism is not true. There must be some other mechanism at play that is causing net transport towards the right.

Related to this point, how is the “swimming force density” calculated, exactly, in the experiments and simulations? I did not see any mention of the precise detailed calculations anywhere (maybe I missed it?). I would imagine that the most direct readout is the polar order around the object, but I do not think the authors have access to this in the experiments. Only the local density is accessible. Are the authors taking spatial gradients of the local density as a proxy of the polar order?

Figure 4A is the telling experiment that further convinces me that the authors' proposed mechanism may have problems. In fact, Figure 4A is confusing and not explained well.

For the two profiles that I boxed in red – I have no idea why the “swimming force density” should change around the overall object (not the colloid, but the overall disk R) when the colloid is inserted into the slow region VS at the interface. Again, the authors do not define how this quantity is computed, so I may be misunderstanding this quantity. Assuming that it is inferred from the local number density, I would expect the bacteria number density to change very locally near the colloid, but the regions around the larger hemi-circle should not change. To me, this result is the most surprising result in the whole paper, and this is not explained at all. Furthermore, this result seems to go directly at odds with the authors' proposed mechanism. If the authors' proposed mechanism were to be true, both of the cases boxed in red should move to the left (not the right) at roughly the same speed. The additional bump created by the colloid at the interface is negligible.

I would like to see a movie of the simulations for both cases boxed in red above.

I propose an alternative physical mechanism to explain the results of this work. The authors should calculate the *total* pressure (active pressure + interaction pressure) around the colloid. Since the crowded, less-active side of the hemi-circle is much more dense compared to the more-active side, the interaction pressures (the usual interaction-based osmotic pressure) are much larger to the left of the colloid. I believe that the *total* pressure is higher on the left side, even though it is less active. The large steric crowded interactions on the left are dominating over the larger active pressure on the right side. The simulation snapshot Fig 2 gives the impression that the crowded side is very dense, possibly close to close packing. The virial pressure shoots to very large values around these packing fractions.

This mechanism would explain the results of Fig 4A, and my point earlier with the red box. When the colloids are embedded into the interiors of the crowded region, since the total pressure is approximately symmetric, there is no net transport. Only when the colloids is place at the interface will the total pressure asymmetry be the largest.

If the authors agree with this alternative mechanism, then it would involve a substantial rewrite and restructuring of the manuscript, since the mechanism is a big component of the paper. I would like to see the force balance worked out by integrating the total pressure around the colloid, and demonstrating fore-aft symmetry breaking. And this “swimming force density” that the authors presented would really not matter anymore.

As I wrote earlier, this review may sound overly critical, but actually I am very supportive of the experiments and the raw results are really meaningful. I’m just worried about the interpretation and the proposed mechanism. In future revisions, I’m very happy to help the authors capture any alternative mechanisms that are more consistent with their data, if they are interested and willing.

Review of “Colloidal transport by light induced gradients of active pressure”

In this work Pellicciotta *et al.* present an experimental and numerical study of passive tracer particles being transported in a bath of bacteria. As they mention, such behavior isn't necessarily new, as it had been previously shown that passive, asymmetric particles can be transported in an homogeneous active bath. However, here the authors reverse the process and show that “the requirement of spatial symmetry breaking [...] can be transferred from the object to the fluid, enabling the active transport of passive objects with arbitrary shapes”.

The authors make a clever use of genetically modified *E. coli* bacteria whose swim velocity can be controlled by light intensity. This way they can easily generate external light fields that directly translate into arbitrary local activity fields. They then show that a passive tracer particle placed at the interface between a low activity and a high activity area is actively pushed towards the later. Notably they show that these observations originate from the alignment of the bacteria at the interface of the activity pattern, which lead to a force imbalance and then transport. These experimental findings are associated with some simple numerical simulations, which highlight the requirement of bacteria interactions (steric alignment in the simulation) to obtain a good agreement with the experiments.

Overall this is a quite novel and well conducted study, which in particular challenges current theoretical predictions of active matter (the passive tracer is expected to move the other way around, towards lower activity areas). As a result I can recommend this work for publication in Nature Communications.

Nonetheless there are several pending questions that I would like the authors to address.

General Questions

1. The authors mention preliminary experiments with smooth activity gradient, but where no significant drift was observed (p.4 l.100). Can the authors rationalize this observation?
2. Similarly, within a light gradient, did the author check for the presence of active torques? Which would affect the bacteria orientation inside the gradient.
3. The author thoroughly investigated the effect of the probe position within the activity pattern, the pattern size itself, and v_0/v_+ the ratio between the global activity of the system v_0 , and the high activity area v_+ . From Fig. 3b and the (quasi) absence of drift speed in the simulations without interactions, the authors conclude that this transport effect originates from an imbalance of the swimming force. However, to my understanding there are no hydrodynamics in the simulations. How is then the swimming force transmitted from the left interface towards the tracer?
4. The authors show that there is an optimal size for the modulation disk radius which maximizes the transport of the tracer. One can expect this length scale to be connected with the transmission of the swimming force from the left interface, to the tracer particle. Naively I would expect the propulsion speed to play a role here. Have the authors checked the effect of v_0 and v_+ on the optimal disk radius?

5. The authors show that optimal transport happens for $V+=V0$ and $v-=0$ (see Fig.3 and SI Fig.4). This eventually results in a passive half disk within an active bath with a tracer at the right interface. If the tracer were put at the left interface, would this result in a reverse (left) drift? Additionally, within this geometry the symmetry is broken since the left circular interface is longer than the right straight one. If the authors were to use a square geometry (and thus symmetrical interfaces), would the transport still happen? I think these inquiries can easily be checked numerically, and could help to unveil the origin of this transport phenomenon.
6. More generally, I believe that the authors could have developed the discussion part a bit more. I understand this is an experimental paper, and the authors actually mention that they hope for the “development of a theoretical framework that allows for at least qualitatively correct predictions”. Nonetheless I would have liked them to provide some hand waved arguments for these controversial (in a positive way) observations!

Specific Remarks

1. l.138: The authors mention steric interactions, probably because they use them in the simulations. Since E. Coli is a pusher, hydrodynamics should also lead to alignment.
2. l.164: I don't see the link between SI Fig.4 and non-vanishing flux of bacteria.
3. l.168: The authors here refer to SI Fig. 3 I believe (not SI Fig. 4).
4. SI p.3: “For mimic” should be “To mimic”.
5. SI p.3: “ $(\alpha_{10})^{-1}$ ” The “10” is probably a typo here.

“Colloidal transport by light induced gradients of active pressure”

Manuscript NCOMMS-22-49098-T

Authors reply

We would like to sincerely thank all reviewers for their important collaboration. Below is a detailed list of responses, where original reviewers' comments appear in *italics* and changes to the manuscript in blue.

Reviewer #1

The manuscript presents experimental data clearly demonstrating that passive colloids suspended in an active fluid can be transported by tailored spatial modulation of the swimming speed of the active agents. Specifically, the authors use light patterns projected on dense suspensions of light-controlled E.coli bacteria to generate areas of high and low swimming centred on passive tracers. Feedback is used to adjust the light pattern (and thus the swimming speed of the bacteria) such that the tracers stay centred on the interface between high and low swimming speed, leading to the observation of a slow systematic drift of the tracer particles towards the side with higher swim speed. The experimentally found direction of drift is unexpected, as simple scaling arguments for an ideal active gas predict that the 'active pressure' exerted by the swimming bacteria should be higher in the region of higher swim speed. The manuscript then presents numerical simulations which show a drift direction consistent with their experimental findings and try to rationalise this in terms of imbalance found in the swimming force density associated with their imposed spatial activity pattern. Although the experimental findings are significant and of general interesting, the link with active (swim) pressure as presented in this manuscript (and advertised in its title) is not convincing. Indeed, after providing a simple scaling arguments that would predict a different direction of motion the authors essentially drop the concept of swim pressure, relying on analysis of swim force density from simulations to add support to their experimental findings. As the authors point out, there is still some ongoing debate about the definition and properties of pressure in active systems and I feel that it is not a useful concept in the context of this experiment.

Reply: We thank Reviewer #1 for considering our work significant and interesting. Besides many technical comments and requests of clarifications the main concern here is that the link with swim pressure is not convincing. This feeling was shared by all other reviewers, and in a way also by ourselves. So we decided to reanalyze our experiments and simulations in more depth, looking for a deeper insight on the role and definition of active pressure in the context of the present experiment. Our new findings lead to a major revision of the manuscript which we believe improves significantly over the original one and we are very grateful to the reviewer for pointing us in this direction.

Below are our responses to these comments.

- 1) **Comment:** *Swim pressure is the pressure needed to confine active particles, i.e. it is analogous of osmotic pressure of passive colloids. While it is straightforward to extract this swim pressure in theoretical treatments by assuming semi-permeable membranes as boundaries, the solid colloid in the scenario presented in this manuscript is not just influenced by the motion of the bacteria but also by the fluid flow set up by the bacteria's motion.*

Reply: We now comment on that point as follows:

Although interactions can also be hydrodynamic in principle, we believe that these play a minor role in our experiments and therefore we do not explicitly consider fluid in our simulations. In fact, hydrodynamic interactions in a force-free system occur only through dipole or higher order multipoles, becoming effective only over short distances where steric interactions are the predominant mechanism for alignment [27]. Moreover, the sample is confined between two closely spaced glass walls which further reduces the range of hydrodynamic interactions [28,29] and ensures that most of the momentum going in the fluid is actually absorbed by the confining walls. Finally, in these experiments bacteria are either highly motile and low concentrated or highly concentrated and non-motile, another reason to justify neglecting of hydrodynamic interactions.

[27] Angelani, L., Di Leonardo, R., Ruocco, G.: Self-starting micromotors in a bacterial bath. *Physical review letters* 102(4), 048104 (2009)

[28] Drescher, K., Dunkel, J., Cisneros, L.H., Ganguly, S., Goldstein, R.E.: Fluid dynamics and noise in bacterial cell–cell and cell–surface scattering. *Proceedings of the National Academy of Sciences* 108(27), 10940–10945 (2011)

[29] Berke, A.P., Turner, L., Berg, H.C., Lauga, E.: Hydrodynamic attraction of swimming microorganisms by surfaces. *Physical Review Letters* 101(3), 038102 (2008)

2) **Comment:** *Although the simulations support the direction of motion found in experiments², I am not sure whether they capture the experimental system sufficiently to provide any real insights. To start with the bacterial densities used in simulations seem extremely high (up to 0.38 area fraction?). Typically an OD = 1 sample of E.coli cells has about 10^9 cells/ml, i.e. a body volume fraction of 0.2%. So even for the densest experimental sample the (3D) volume fraction would have been below 5%. Could the authors comment?*

Reply: We agree that a discussion was needed here to justify the estimates given in the main text, we now clarify this point in the SI Section 2 (Numerical model):

In the experiments, the cell area packing fraction ϕ can be estimated from the cell concentration (measured in optical density units) and considering that most of the bacteria accumulate at the bottom and top surfaces (persistence length greater than the sample cell height) [4].

Using our cell growth protocol, we have that the cell concentration is $N_c = 10^9$ cells/ml for an optical density OD=1 [5], and cells with an average length $l = 2.6 \pm 0.6 \mu\text{m}$ and diameter $\sigma = 0.86 \pm 0.07 \mu\text{m}$. Assuming that half of the bacteria in the sample accumulate at the bottom surface, the number of cells on a surface of $1 \mu\text{m}^2$ area is $N_b = N_c h/2$, where $h = 20 \mu\text{m}$ is the chamber height. Then, the area packing fraction ϕ is the number of cells at the bottom times the cross-sectional area of a bacterium, $\phi = N_b l \sigma \approx 0.02$ for OD =1. We found that the best quantitative agreement between simulations and experiments is reached when we consider $\phi = 0.028$, which is not too far from the estimated value considering the large deviations in the cell size.

[4] Berke, A.P., Turner, L., Berg, H.C., Lauga, E.: Hydrodynamic attraction of swimming microorganisms by surfaces. *Physical Review Letters* 101(3), 038102 (2008)

[5] Schwarz-Linek, J., Arlt, J., Jepsen, A., Dawson, A., Vissers, T., Miroli, D., Pilizota, T., Martinez, V.A., Poon, W.C.: Escherichia coli as a model active colloid: A practical introduction. *Colloids and Surfaces B: Biointerfaces* 137, 2–16 (2016)

3) **Comment:** *And the snapshot shown in fig.2A suggests that the low intensity region has an even higher density which seems to lead to local alignment and presumably correlated motion. Surely hydrodynamic interactions between swimmers (and the colloid) can no longer be neglected?*

Reply: See reply to point 1)

- 4) **Comment:** *Fig. 3C also suggests that the drift speed is not so much controlled by the velocity gradient at the interface next to the tracer colloid, but rather by the (curved) interface of the low intensity region. Could the authors comment more why this might be the case?*

Reply: This is now discuss in depth in the revised manuscript and summarized in the conclusions as follows:

In particular we find that when a probe particle is placed at the interface between a slow and a fast region, although swim pressure alone should be higher on the fast side, a layer of polarized cells surrounding the slow area generates there a direct pressure component that is strong enough to push the particle towards the fast, higher swim pressure side.

- 5) **Comment:** *I assume the density in the dark region will be very different for varying 'background' illumination?*

Reply: That is correct and we now address this point in the new Fig.4c, Supplementary Figure 4, and corresponding text in the main:

It should be noted however, that although the speeds in the slow and fast half disks remain constant, changing the background speed v_0 results in density readjustments as shown in Fig.4c. Ideally, for non-interacting bacteria, density ratios should match speed ratios with the constraint of conserving the total cell number. Specifically, the densities ρ_+ and ρ_- , when normalized to the background density ρ_0 , should increase linearly with v_0 with slopes given by $1/v_+$ and $1/v_-$, respectively, shown as solid lines in Fig.4c. In the dark region, however, ρ_- follows this ideal trend only up to a saturation value that is approximately twice the background density ρ_0 and where steric interactions probably prevent further accumulation. Both this saturation of ρ_- and the associated slowing down of bacteria in densely packed regions should decrease swim pressure on the slow side [7] while direct pressure shoots up due to close packing resulting in a faster drift.

[7] Solon, A.P., Stenhammar, J., Wittkowski, R., Kardar, M., Kafri, Y., Cates, M.E., Tailleur, J.: Pressure and phase equilibria in interacting active brownian spheres. *Physical review letters* 114(19), 198301 (2015)

- 6) **Comment:** *The experimental evidence provided in the manuscript is of general interest and seems methodologically sound, but the manuscript would benefit*

from some more detailed characterisation of their experimental system: E. coli strain. It is known that light-controlled bacteria do not react instantly to changes in light intensity. Seeing that bacteria only need a few seconds to traverse different regions even a small delay could have a large experimental impact. Furthermore, on a similar light-control run and tumble strain it has been found that their ‘tumbling rate increases noticeably as cells swim from light to dark, whereas cells swimming from dark to light do not show any obvious change in their tumbling behaviour.’ (J.Art et al., Nat Comm. 10, 1 (2019)), which could potentially complicate the picture further. Is this also the case here?

Reply: Yes, this happens in this case as well. However, the scaling of the density in the new Fig.4c and the general agreement between the experimental data and the numerical simulations (in which both of these effects are neglected) reassure us of the robustness of our results and the interpretation we propose with respect to these two and many other neglected effects (hydrodynamics, heterogeneous motility, boundary effects, non-uniformity and blurring of the projected patterns, etc.).

- 7) **Comment:** *Density profile: Fig. 1 shows that there is a clearly defined high density region where the light intensity is lower. How does this density compare to a) the initial density, b) the density in the high intensity and background intensity region? Visually it looks much denser than would be expected if $\rho \cdot v$ where constant (which of course is unlikely to apply in this experiment)*

Reply: As discussed in the reply to point 5), we extracted the densities from the experiments and studied how they change as the background illumination changes. Interestingly, the product $\rho \cdot v$ remains fairly constant when we move across regions of not too high densities. We now discuss this in the new Fig. 4c, and also in the Supplementary Figure 4).

- 8) *... And can you observe correlated bacterial motion in these high density regions?*

Reply: Unfortunately, because of the low magnification used for this experiment, we were unable to measure whether correlated motion is present in the dark region.

- 9) **Comment:** *How long does it take for such a dense region to be established? The manuscript suggests that the colloids only start moving ‘systematically’ after this*

has been established. Does it (on average) stay centred on the interface during this time? And how long does it take the particle to reverse direction after the direction of the pattern has been flipped?

Reply: This is an interesting detail that deserved to be discussed. We now included two new figures in the Supplementary Figure 3 reporting density and drift speed dynamics during the transient. The following discussion is included as a caption of this new figure:

Establishment of bacteria density and drift speed during light pattern projection. (a) Cell density in the dark region (I_{-}) during the first 50s of light pattern projection and starting from a homogeneous concentration. The density is normalized by its initial value. (b) The corresponding drift speed of the particle in the first 50s. The particles start to move as soon as the density in the darker region increases. When the density reaches its maximum, the particle velocity seems also to fluctuate around a constant value. By fitting the two curves with an exponential ($y = A(1 - e^{-t/\tau}) + C$), we found a characteristic time τ for the density and velocity establishment of 5 and 7 seconds, respectively. The fits are shown as green dashed lines. After reaching a maximum, the density slowly decreases due to the overall decrease of bacteria from the field of view (caused by the background illumination). Flipping the direction of colloidal transport would require restoring a homogeneous bacteria density around the particle, then flipping the pattern direction and waiting for bacteria pattern formation. Since restoring a homogeneous density around the particles takes only a few seconds, the time of flipping direction is mainly ruled by the transient time τ . Each data point is an average of more than 50 particles, error bars are the standard errors of the mean.

10) **Comment:** *Some of the authors have recently demonstrated how the swimming of light controlled bacteria can be rectified by an optical feedback loop (Nat. Comm 13, 1 (2022)). Have the authors considered that something similar might be happening here? E.g. the dense region might follow shifts of the pattern towards the brighter half more quickly than shifts in the opposite direction.*

Reply: The Reviewer's is suggesting here an interesting alternative interpretation: what if drift is just a consequence of random bead fluctuations being rectified by an "asymmetric mobility" of the dense region? This scenario assumes that bacteria in the dense region translate and drag the bead along their motion. However, as we report in Supplementary Figure 7, the average speed of bacteria in the slow region is 4 times smaller than the bead's drift

speed. We also performed simulations with a fixed bead and found a net force pointing to the same direction of the observed drift. Therefore our simulations do not support this alternative interpretation.

11) **Minor points/corrections:**

a) *Scale bar size in fig. 1b&c not mentioned.*

Reply: Corrected

b) *Eq (1) v_i on LHS should be typeset as vector as well.*

Reply: Corrected

c) *Caption fig 2A): Pink and blue bacteria are those swimming toward the regions with higher and lower activity, respectively. Do you really mean that? Not simply pink/blue to positive/negative x direction?*

Reply: Yes we mean positive/negative x, caption corrected.

d) *Fig. 3. A) where are the boundaries of the light region? I.e. is the pink region on the LHS just inside or outside the dark area?*

B) Scale would be useful: at least centre position in x and 0 in y

Reply: Fig.3A and 3B do not appear in the revised manuscript

e) *Line 179/180: shouldn't this read 'for the integrated positive contribution'?*

Reply: discussion removed.

f) *Caption supp fig. 1: Hill equation expression wrong.*

Reply: Corrected

g) *Caption supp fig 2. Orange line (not blue line)*

Reply: Corrected

Reviewer #2

In this manuscript, Pellicciotta et al use light-sensitive motile bacteria to produce spatial variations in bacterial activity. They show that a passive colloid placed in a bacterial bath with spatial variations in activity can achieve net directed transport. The authors write that it is surprising that the colloids move from regions of low to high activity based on arguments of the ideal-gas pressure. Interactions between the swimmers are critical to the observed phenomenon, indicating that the ideal-gas like pressure arguments alone fail to explain the results. Apparently, the crowded bacteria on the less active side act as a connected object that gets pushed collectively from the bacteria in the bulk (background) region. To validate this hypothesis, the authors vary the background activity, and the colloidal transport is maximized at the largest background activity (Fig. 3). Surprisingly, Figure 4 shows that the localization of the colloid is critical – the colloid placed at the interface produces the largest transport, whereas the colloid placed within the interiors of the disk does not.

The experiments are very nice; very creative use of light-induced bacterial activity. Results are also interesting. I am willing to reconsider a revised version of this manuscript, but in the current manuscript, as written, I have a concern with the proposed mechanism and model for the observed phenomena. I do not have an issue with the results/data, but I have an issue with the authors' interpretation and mechanism.

This review may sound overly critical, but I am actually quite supportive. The experiments are very nice and will add to the body of work in a meaningful way. I genuinely want to help the authors improve this work. I just disagree with the proposed mechanism, and this may require a significant reconstruction of the manuscript.

Reply: We thank Reviewer #2 for appreciating our work and for their support. Above all, we thank the reviewer for stimulating us, with a long series of criticisms and suggestions, to radically rethink the interpretation of our data. Based on their suggestion, we used the simulation results to calculate interaction pressure directly and swimming pressure indirectly. The picture that emerges fully confirms the reviewer's intuition: the drift originates mainly from virial pressure that "shoots up" due to compression exerted by a layer of polarized cells surrounding the slow region. We restructured completely the manuscript sections containing the interpretation of the experimental results. All texts and figures that had raised the reviewer's concerns have been replaced according to the new interpretation. For this reason, there is no point in proceeding with a detailed list of responses, but we simply invite the reviewer to read the revised manuscript in the hope that this time they will appreciate the physical insight it contains along with the experimental achievements.

Here follows just a quick summary of clarifications and changes made in response to the the reviewer's comments:

- 1) **Comment:** *First, I believe that the way the manuscript is introduced is misleading and unrelated to the major conclusions of this work. In the abstract, the authors bring up the ideal-gas active pressure, and that the probe moves from regions of low to high pressure. This is an argument based on dilute, non-interacting systems. However, the main results of this work is entirely reliant upon nonideal interactions between the bacteria. I understand the need to be provocative, especially for high- impact journals like this, but it's to a degree and at the expense of scientifically misrepresenting the problem. After reading the abstract and the introduction, I had the impression that the bacteria are dilute, and two-body interactions between the bacteria plays no role in this "surprising" result. In reality, the interactions play an essential role in the transport. I still believe that the colloid is going from high to low total pressure, which includes the active pressure and interaction pressures. So the results are actually not that surprising to me, when presented in this way.*

Reply: We rewrote the introduction to include from the beginning a discussion on the role of interactions. A little "surprise effect" is still present, albeit in a more mitigated form, since even in the simulations now cited as ref [14], although the direct pressure contribution almost completely counterbalances the swim pressure, the total pressure remains slightly higher in the more active half.

- 2) **Comment:** *According to the authors' proposed mechanism, the whole assembly (colloid + dense bacteria on the less-active hemi-circle) is being transported, as opposed to moving just the colloid. Presumably, one could just place a rigid object in the shape of a hemi-circle, and the hemi-circle would still move. More generally, one could just create any object with sufficient asymmetry and curvature, and the object should move. This fact seems to decrease the impact of this manuscript a little bit, if this mechanism is really true. If I understood correctly, the authors' proposed mechanism is that effectively the system takes on the shape that looks like the following:*

The authors claim that this background activity breaks the symmetry of forces and causes net translation of the object as a whole. I disagree with this proposed mechanism. The arguments presented in Eqs (2) – (4) are weak in my opinion. A better approach would be to follow Yan & Brady “The force on a boundary in active matter”, JFM 2015, which is not cited in the current manuscript (to me, this is an important missing reference, given the authors’ proposed mechanism). Equation (2.2) in Yan & Brady is the direct calculation of the force on any object, given the number density of particles. In short, this expression states that every particle gives a kT -kick on the boundary, so all we need to do is to integrate the number density of particles around the whole boundary surface. I encourage the authors to calculate this force based on the known density at every point around this whole object in both the experiments and simulations, shown above. I doubt that the authors will calculate a net positive force towards the right. Yan & Brady 2015 JFM showed that a strong concave curvature is required to achieve sufficient asymmetry in kT -kicks for net directed transport of the whole object. See Fig. 3B in Yan & Brady 2015 JFM – their shape almost looks like the shape drawn above (!), except without the concave curvature. Furthermore, this would argue that the object would go in the opposite direction, towards the left! I find it hard to believe that the authors’ presented mechanism would cause the object above to achieve net directed transport towards the right. For me to believe the authors’ proposed mechanism, I would like to see a simulation of a rigid object in the shape of the whole disk object (the shape I drew above), placed inside a bath of active particles swimming at the bulk background activity. If the simulations show that the whole object moves towards the right, I would be convinced (I would be very surprised if this happens!). The only thing I can think of is the shape of the active particles are non-spherical, unlike the spherical particles used in the theory – but I doubt that the small deformation in active particle shape would reverse the direction of net transport. Of course, the experiments are what they are, and they show that the object moves towards the right. I conclude then that the authors’ proposed mechanism is not true. There must be some other mechanism at play that is causing net transport towards the right.

Reply: Perhaps we were misunderstood in the earlier version of the manuscript. We were aware that the dense half does not move rigidly with the bead (the two speeds are very different, see also the answer to Reviewer #1's point 10). The main argument, which by the way remains true even though, as the reviewer points out, it does not offer a clear physical interpretation, was that there is an overall force balance and that the integrated swimming force of the bacteria must equal the total force on the bead plus any current in the bacteria. However, this part has been removed from the revised version.

- 3) **Comment:** *Related to this point, how is the “swimming force density” calculated, exactly, in the experiments and simulations? I did not see any mention of the precise detailed calculations anywhere (maybe I missed it?). I would imagine that the most direct readout is the polar order around the object, but I do not think the authors have access to this in the experiments. Only the local density is accessible. Are the authors taking spatial gradients of the local density as a proxy of the polar order?*

Reply: The swim force density is only accessible from simulations, we now state this clearly in the main text and provide a definition in the Supplementary Information - 2. Numerical Model.

- 4) **Comment:** *Figure 4A is the telling experiment that further convinces me that the authors’ proposed mechanism may have problems. In fact, Figure 4A is confusing and not explained well. For the two profiles that I boxed in red – I have no idea why the “swimming force density” should change around the overall object (not the colloid, but the overall disk R) when the colloid is inserted into the*

slow region VS at the interface. Again, the authors do not define how this quantity is computed, so I may be misunderstanding this quantity. Assuming that it is inferred from the local number density, I would expect the bacteria number density to change very locally near the colloid, but the regions around the larger hemi-circle should not change. To me, this result is the most surprising result in the whole paper, and this is not explained at

all. Furthermore, this result seems to go directly at odds with the authors' proposed mechanism. If the authors' proposed mechanism were to be true, both of the cases boxed in red should move to the left (not the right) at roughly the same speed. The additional bump created by the colloid at the interface is negligible. I would like to see a movie of the simulations for both cases boxed in red above.

Reply: Fig.4a, and relative discussion, has been removed. However, we would like to point out that only when the bead is at the interface does the increased pressure from the slow side cause a net drift, whereas the system is stationary when the bead is fully immersed in either the slow or the fast side. So, what causes a different arrangement of bacteria around the larger hemicycle is the fact that in one case (central figure in the top row of old Fig.4A) the system is dynamic and drifting while in the other two cases the system is stationary but with zero currents. The mechanism we proposed was not that of an effective rigid 'dressed' bead push by bacteria outside, but that of an emerging net swim force density induced by an asymmetric activity pattern. We now realise that, although still true, this was not an explanation, but rather a fact. The interpretation we offer now, inspired by the reviewer's intuition, provides the physical insight we (authors and reviewers) were all looking for.

- 5) **Comment:** *I propose an alternative physical mechanism to explain the results of this work. The authors should calculate the total pressure (active pressure + interaction pressure) around the colloid. Since the crowded, less-active side of the hemi-circle is much more dense compared to the more-active side, the interaction pressures (the usual interaction-based osmotic pressure) are much larger to the left of the colloid. I believe that the total pressure is higher on the left side, even though it is less active. The large steric crowded interactions on the left are dominating over the larger active pressure on the right side. The simulation snapshot Fig 2 gives the impression that the crowded side is very dense, possibly close to close packing. The virial pressure shoots to very large values around these packing fractions. This mechanism would explain the results of Fig 4A, and my point earlier with the red box. When the colloids are embedded into the interiors of the crowded region, since the total pressure is approximately symmetric, there is no net transport. Only when the colloids is place at the interface will the total pressure asymmetry be the largest. If the authors agree with this alternative mechanism, then it would involve a substantial rewrite and restructuring of the manuscript, since the mechanism is a big component of the paper. I would like to see the force balance worked out by integrating the total pressure around the colloid, and demonstrating fore-aft symmetry breaking. And this "swimming force density" that the authors presented would really not matter*

anymore. As I wrote earlier, this review may sound overly critical, but actually I am very supportive of the experiments and the raw results are really meaningful. I'm just worried about the interpretation and the proposed mechanism. In future revisions, I'm very happy to help the authors capture any alternative mechanisms that are more consistent with their data, if they are interested and willing.

Reply: Once again, we would like to thank the Reviewer for pushing us in a direction that has finally led to the formulation of an interpretative framework that is now clear and consistent within the mechanics of active systems. The manuscript was substantially restructured and rewritten, and the discussion was enriched with new data and figures to support the proposed new mechanism.

Reviewer #3

In this work Pellicciotta et al. present an experimental and numerical study of passive tracer particles being transported in a bath of bacteria. As they mention, such behavior isn't necessarily new, as it had been previously shown that passive, asymmetric particles can be transported in an homogeneous active bath. However, here the authors reverse the process and show that "the requirement of spatial symmetry breaking [...] can be transferred from the object to the fluid, enabling the active transport of passive objects with arbitrary shapes".

The authors make a clever use of genetically modified E. coli bacteria whose swim velocity can be controlled by light intensity. This way they can easily generate external light fields that directly translate into arbitrary local activity fields. They then show that a passive tracer particle placed at the interface between a low activity and a high activity area is actively pushed towards the later. Notably they show that these observations originate from the alignment of the bacteria at the interface of the activity pattern, which lead to a force imbalance and then transport. These experimental findings are associated with some simple numerical simulations, which highlight the requirement of bacteria interactions (steric alignment in the simulation) to obtain a good agreement with the experiments. Overall this is a quite novel and well conducted study, which in particular challenges current theoretical predictions of active matter (the passive tracer is expected to move the other way around, towards lower activity areas). As a result I can recommend this work for publication in Nature Communications. Nonetheless there are several pending questions that I would like the authors to address.

Reply: We thank the Reviewer for appreciating the quality of our work and novelty of our results. The Reviewer recommends publication after some pending questions are resolved.

General Questions

- 1) **Comment:** *The authors mention preliminary experiments with smooth activity gradient, but where no significant drift was observed (p.4 l.100). Can the authors rationalize this observation?*

Reply: We realized that this observation needed backup data and a better explanation. We now report in the Supplementary Figure 2 the trajectories and mean speeds of tracer particles in smooth activity gradients. Within the new interpretation framework, we can now rationalize this observation as follows:

It is also interesting to compare this estimated value of p_S to the swim pressure $\gamma \rho v_+^2 \tau / 2$ of an ideal active gas of run and tumble particles in 2D. Using simulation parameters (in reduced units as defined in the Supplementary Information) $\gamma = 1$, $v_+ = 1$, $\tau = 10$ (inverse tumbling rate) and the average value of the particle density in the fast region $\rho = 1$, we find an ideal swim pressure that is greater than p_S by an order of magnitude. This may be due to an effective interaction time τ that is not controlled by the inverse tumbling rate but rather by the typical interaction time with the bead scaling with $a/v_+ = 2.5$. We can speculate that an interaction time that scales as the inverse speed may also be the reason why the swim pressure $\sim \rho v_+^2 \tau$ is practically constant ($\rho \propto 1/v_+$, $\tau \propto 1/v_+$) in the absence of interactions (see Fig.2c) or at low densities such as those obtained in smooth speed gradients (see Supplementary Figure XX).

- 2) **Comment:** *Similarly, within a light gradient, did the author check for the presence of active torques? Which would affect the bacteria orientation inside the gradient.*

Reply: No active torque is expected on bacteria even when exposed to light gradients. Light is in fact absorbed by proteorhodopsin to generate a proton motive force that quickly equilibrates to be constant across the entire cell membrane to power flagellar motors equally. An interesting related point is whether an active torque can be generated on anisotropic beads. We now include a discussion on that as a perspective.

Future studies may address the possibility of generating a net torque on

anisotropic and achiral objects suspended in bacterial baths with spatially patterned motility.

- 3) **Comment:** *The author thoroughly investigated the effect of the probe position within the activity pattern, the pattern size itself, and v_0/v_+ the ratio between the global activity of the system v_0 , and the high activity area v_+ . From Fig. 3b and the (quasi) absence of drift speed in the simulations without interactions, the authors conclude that this transport effect originates from an imbalance of the swimming force. However, to my understanding there are no hydrodynamics in the simulations. How is then the swimming force transmitted from the left interface towards the tracer?*

Reply: We now propose a different mechanism for the drift where steric (not hydrodynamic) interactions are the main responsible for the appearance of a direct pressure component that pushes the particle from the slow side. The whole manuscript is now restructured around this idea and the role of steric interactions in transmitting the force exerted by polarized cells should be clearly stated.

- 4) **Comment:** *The authors show that there is an optimal size for the modulation disk radius which maximizes the transport of the tracer. One can expect this length scale to be connected with the transmission of the swimming force from the left interface, to the tracer particle. Naively I would expect the propulsion speed to play a role here. Have the authors checked the effect of v_0 and v_+ on the optimal disk radius?*

Reply: We thank the reviewer for pushing us in exploring this aspect in greater detail. As shown in Supplementary Fig.6b the optimal disk radius depends only weakly on the speeds v_0 , v_+ .

- 5) **Comment:** *The authors show that optimal transport happens for $V_+=V_0$ and $v_-=0$ (see Fig.3 and SI Fig.4). This eventually results in a passive half disk within an active bath with a tracer at the right interface. If the tracer were put at the left interface, would this result in a reverse (left) drift?*

Reply: This is a very interesting point that needed to be checked. We find that the speed is actually reversed as shown in the inset of new Fig.3c.

- 6) **Comment:** *Additionally, within this geometry the symmetry is broken since the left circular interface is longer than the right straight one. If the authors were to use a square geometry (and thus symmetrical interfaces), would the transport still happen? I think these inquiries can easily be checked numerically, and could help to unveil the origin of this transport phenomenon.*

Reply: Within the new interpretation framework, what breaks the symmetry is that only half of the bead is immersed in a slow region that is compressed by a layer of polarized cells over its boundary. This is still the case for a more “symmetric” interface like a square as now demonstrated by new simulation data reported in Supplementary Figure 5. We now comment on that point in the revised manuscript:

Stated this way, the drift should not be strongly influenced by the shape of the modulation boundary and indeed this is confirmed by simulations using a square modulation pattern with the same area as the disk (see Supplementary Fig. 5). However, changing the size of the modulation area can have strong effects on the drift speed, affecting the density ratios in the three regions with different activities (see Supplementary Fig. 6).

- 7) **Comment:** *More generally, I believe that the authors could have developed the discussion part a bit more. I understand this is an experimental paper, and the authors actually mention that they hope for the “development of a theoretical framework that allows for at least qualitatively correct predictions”. Nonetheless I would have liked them to provide some hand waved arguments for these controversial (in a positive way) observations!*

Reply: We thank the Reviewer for prompting us to find a more satisfactory physical picture for our results. Following their advice, we believe we are now resubmitting a completely restructured manuscript that overcomes many of the interpretative limitations of the original version.

Specific Remarks

1. I.138: *The authors mention steric interactions, probably because they use them in the simulations. Since E. Coli is a pusher, hydrodynamics should also lead to alignment.*

Reply: We now include a better justification of why hydrodynamic interactions should not play a primary role here (see reply to comment 1 of Reviewer #1).

2. *I.164: I don't see the link between SI Fig.4 and non-vanishing flux of bacteria.*

Reply: No longer relevant (figure removed)

3. *I.168: The authors here refer to SI Fig. 3 I believe (not SI Fig. 4).*

Reply: Corrected

4. *SI p.3: "For mimic" should be "To mimic".*

Reply: Corrected

5. *SI p.3: " $(\alpha_{10})^{-1}$ " The "10" is probably a typo here.*

Reply: Corrected

REVIEWERS' COMMENTS

Reviewer #1 (Remarks to the Author):

The authors extensively modified their original manuscript to address the various points raised by all of the reviewers. They have re-analysed their simulations to provide a far more convincing explanation for their experimental findings. Overall this revised manuscript is significantly improved and should now be suitable for publication in Nature Communications. Overall the authors have addressed all the points the referees have raised in a very detailed and thorough manner and updated their manuscript accordingly. However, I feel that they almost have 'overdone' their reply to my queries regarding the role of hydrodynamics, adding 4 long sentences at the bottom of page 6. The argument presented in the 3rd sentence is not logically consistent (e.g. citing closeness to the boundary as reason for a limited hydrodynamic range despite the surface accumulation being induced by hydrodynamic interactions in the first place), so I would suggest to drop the last 2 sentences (and maybe instead mention other effects that also had to be ignored to keep the simulations manageable/instructive).

Another small point: In response to my comment 11 the authors state that they '... also performed simulations with a fixed bead and found a net force pointing to the same direction of the observed drift to rule out that the observed motion is an artefact of the feedback mechanism. This might be worthwhile mentioning somewhere in the SI.

Reviewer #2 (Remarks to the Author):

I celebrate the author's efforts to make significant changes to their manuscript. In particular, the interpretation of the mechanisms of the main results have changed completely. Their interpretation is now consistent with my intuition of the system and I have no further comments to the work. I am very excited to see this manuscript published in this journal. Thank you for the opportunity to review this manuscript.

Reviewer #3 (Remarks to the Author):

In my opinion the originally submitted version of this work was already quite interesting, both in terms of quality and novelty of the experimental results.

My only issue was with the suggested mechanism for this reverse motion, an issue that was shared by the other reviewers.

The amount of work spent by the authors to completely rewrite the theory is quite impressive, and the resulting discussion of the mechanism is now much more convincing. As a result, the authors have satisfactorily addressed all my concerns, and I am happy to recommend this work for publication.

Reviewer #1

Comment:

The authors extensively modified their original manuscript to address the various points raised by all of the reviewers. They have re-analysed their simulations to provide a far more convincing explanation for their experimental findings. Overall this revised manuscript is significantly improved and should now be suitable for publication in Nature Communications. Overall the authors have addressed all the points the referees have raised in a very detailed and thorough manner and updated their manuscript accordingly. However, I feel that they almost have 'overdone' their reply to my queries regarding the role of hydrodynamics, adding 4 long sentences at the bottom of page 6. The argument presented in the 3rd sentence is not logically consistent (e.g. citing closeness to the boundary as reason for a limited hydrodynamic range despite the surface accumulation being induced by hydrodynamic interactions in the first place), so I would suggest to drop the last 2 sentences (and maybe instead mention other effects that also had to be ignored to keep the simulations manageable/instructive).

Reply:

We thank the reviewer for recommending our work for publication. Although we believe the argument is still logically consistent (cell-wall hydrodynamic interactions are strong and contribute to wall accumulation while cell-cell hydrodynamic interactions are screened by image singularities induced by the no-slip wall), we have removed the two sentences to avoid misunderstandings.

Comment:

Another small point: In response to my comment 11 the authors state that they '... also performed simulations with a fixed bead and found a net force pointing to the same direction of the observed drift to rule out that the observed motion is an artefact of the feedback mechanism. This might worthwhile mentioning somewhere in the SI.

Reply:

We thank the Reviewer for pointing this missing information. The new Figure 9 in Supplementary Materials now shows simulation data with a fixed bead.

Reviewer #2

Comment:

I celebrate the author's efforts to make significant changes to their manuscript. In particular, the interpretation of the mechanisms of the main results have changed completely. Their interpretation is now consistent with my intuition of the system and I have no further comments to the work. I am very excited to see this manuscript published in this journal. Thank you for the opportunity to review this manuscript.

Reply:

We thank the reviewer for recommending our work for publication.

Reviewer #3

Comment:

In my opinion the originally submitted version of this work was already quite interesting, both in terms of quality and novelty of the experimental results.

My only issue was with the suggested mechanism for this reverse motion, an issue that was shared by the other reviewers.

The amount of work spent by the authors to completely rewrite the theory is quite impressive, and the resulting discussion of the mechanism is now much more convincing. As a result, the authors have satisfactorily addressed all my concerns, and I am happy to recommend this work for publication.

Reply:

We thank the reviewer for recommending our work for publication.